# BENCHMARKING AND ADVANCING QUANTIZATION-AWARE TRAINING FOR REASONING MODELS

## ABSTRACT

Reasoning models have excelled at complex tasks such as coding and mathematical competitions, yet their reasoning processes suffer from low inference efficiency. Quantization is a popular way to boost efficiency, but prior work shows that it causes large performance drops in these models. To address this, we comprehensively benchmark the quantization-aware training (QAT) for reasoning models. Our key findings are: (1) knowledge distillation serves as a versatile objective for reasoning models trained with either supervised fine-tuning or reinforcement-learning algorithms; (2) post-training quantization (PTQ) provides a strong initialization for QAT, improving accuracy while reducing training cost; (3) QAT with reinforcement learning is feasible and yields additional gains for the quantized model; and (4) aligning the domain of QAT training data with the PTQ calibration data further improves the performance. Building on these insights, we propose Reasoning-QAT, an optimized QAT workflow tailored to reasoning models. Empirical results show that Reasoning-QAT outperforms state-of-the-art PTQ methods across multiple LLM backbones and reasoning datasets. For instance, on the DeepSeek-R1-Qwen-Distill-1.5B model, Reasoning-QAT surpasses FlatQuant by 2.92% under W4A4KV4 quantization and GPTQ by 4.74% under W3G128 quantization, respectively.

## 1 INTRODUCTION

Recent large language models (LLMs) (Jaech et al., 2024; Guo et al., 2025; Team et al., 2025) endowed with enhanced reasoning capabilities, have achieved remarkable progress in domains like coding and mathematics. However, these advances come with a trade-off: reasoning-focused inference is often slower and more redundant, resulting in huge inference overhead (Qu et al., 2025). Quantization has emerged as a promising technique to accelerate LLM inference (Frantar et al., 2022a; Lin et al., 2023; Li et al., 2024; Liu et al., 2024; Li et al., 2025a). However, prior studies (Li et al., 2025b; Srivastava et al., 2025; Liu et al., 2025a) have shown that post-training quantization (PTQ) can cause significant performance degradation in reasoning models under extreme-low bit scenarios, such as 3-bit weight-only quantization or 4-bit weight–activation quantization. We corroborate this finding by comparing quantized LLMs on both non-reasoning and reasoning tasks (see Figure 1). As observed, the 4-bit weight quantization with the group-size 128 achieves near-lossless results across tasks, whereas 3-bit variants suffer large performance drops. Note that this degradation is much more severe in reasoning tasks than in non-reasoning ones.

To address this limitation, quantization-aware training (QAT) (Tailor et al., 2020; Nagel et al., 2022; Liu et al., 2023; Bondarenko et al., 2024) provides an appealing alternative by explicitly simulating low-precision inference during training. While QAT has demonstrated effectiveness for general-purpose LLMs (Liu et al., 2023; Chen et al., 2024), it remains unclear whether such benefits can be extended to reasoning models. Several unique challenges for applying QAT to reasoning-focused LLMs arise in this context: (1) the uncertainty of which QAT objective is most suitable for continual training in reasoning models; (2) the prohibitive training overhead that limits practical deployment; and (3) the lack of consensus on how to select effective QAT training data.

In this study, we present a comprehensive benchmark of quantization-aware training (QAT) for reasoning models. We investigate two representative low-bit settings: 3-bit weight-only quantization with a group-size 128 and 4-bit weight–activation quantization. Our benchmark covers mod-

els trained under two major reasoning paradigms: (i) supervised fine-tuning (SFT), represented by DeepSeek-R1-Qwen-Distill-1.5B (Guo et al., 2025); and (ii) reinforcement learning (RL), represented by Qwen3-0.6B and Qwen3-4B (Yang et al., 2025). The evaluation is conducted on a wide range of reasoning benchmarks, including AIME-120, MATH-500 (Lightman et al., 2023), GSM8K (Cobbe et al., 2021), GPQA-Diamond (Rein et al., 2024), and LivceCodeBench (Jain et al., 2024). Our empirical findings are summarized as follows: (1) Knowledge distillation (KD) (Hinton et al., 2015) proves to be a powerful training objective, applicable to reasoning models trained via both SFT and RL; (2) PTQ can serve as a strong initialization for QAT, enabling higher accuracy while reducing training costs; (3) QAT combined with RL is not only feasible but also delivers further improvements to quantized models; (4) it is preferred to keep the QAT training dataset and PTQ calibration data from the same domain, which empirically benefits the quantized reasoning models.

Building on these insights, we propose Reasoning-QAT, an optimized QAT workflow tailored to reasoning-focused LLMs. Reasoning-QAT incorporates three steps: (1) PTQ-based initialization, with the purpose of enhance the model's tolerance to quantization noise and provide a better starting point for subsequent training; (2) knowledge distillation, which aligns the quantized model's output distribution with that of its full-precision counterpart. It recovers the performance and paves the way for the next stage; and (3) cold-start RL, which adopts GRPO (Guo et al., 2025) as the RL paradigm on top of the knowledge-distilled model. This stage reduces entropy and enforces more deterministic and reliable outputs. Extensive experiments demonstrate that Reasoning-QAT consistently outperforms the state-of-the-art PTQ methods across multiple LLM backbones and reasoning benchmarks. For example, on the DeepSeek-R1-Qwen-Distill-1.5B model, Reasoning-QAT can surpass FlatQuant (Sun et al., 2024) by **2.92**% under W4A4KV4 quantization and GPTQ (Frantar et al., 2022a) by **4.74**% under W3G128 quantization.

To the best of our knowledge, this is the first comprehensive benchmark of quantization-aware training on reasoning models. We hope our research provides valuable guidance for the community toward better quantization methods for reasoning models.

## 2 PRELIMINARIES

### 2.1 POST-TRAINING QUANTIZATION FOR REASONING MODELS

**Background and Notations.** Quantization has been a popular approach for the compression and acceleration of LLMs. Given the model parameters $\mathbf{W}$ stored in the bfloat16 format, quantization aims to convert it to low bit representations $\mathbf{W}_{int}$, i.e.,

$$\mathbf{W}_{int} = \text{clip}(\lfloor \frac{\mathbf{W}}{s} \rceil + z, Q_{min}, Q_{max}), \tag{1}$$

where $\text{clip}(\cdot, Q_{min}, Q_{max})$ truncates the associate values inside the minimal $Q_{min}$ and maximal $Q_{max}$, $s$ is the scaling factor and $z$ is the zero point. For $N$-bit symmetric quantization, $s = \frac{\max(|\mathbf{W}|)}{2^{N-1}-1}$ and $z = 0$. For asymmetric quantization, $s = \frac{max(\mathbf{W})-min(\mathbf{W})}{2^N-1}$, $z = \lfloor \frac{-min(\mathbf{W})}{s} \rceil$.

For weight quantization, the low bit quantized weights $\mathbf{W}_{int}$ in the forward pass are then dequantized to $\hat{\mathbf{W}} = s \cdot (\mathbf{W_{int}} - z)$ for the following operations. Instead, for weight-activation quantization, both quantized weights $\mathbf{W}_{int}$ and activations $\mathbf{X}_{int}$ are stored as low bit integers, and their integer matrix multiplication kernel can further reduces the computation aside from size reduction.

**Post-training quantization incurs a large performance drop on reasoning models.** A majority work of LLM quantization focuses on post-training quantization (PTQ) (Frantar et al., 2022b; Lin et al., 2023; Ashkboos et al., 2024; Sun et al., 2024), where the model is directly quantized without training. PTQ is usually fast and easy to implement, with satisfactory performance on most general natural language tasks. However, recent studies (Liu et al., 2025a) show that quantized reasoning models still exhibit large performance drops particularly on challenging tasks.

To further validate this, we compare the quantized LLMs with PTQ methods on both non-reasoning and reasoning models. From Figure 1, it can be found that for DeepSeek-R1-Distill-Qwen-1.5B (abbr. R1-Qwen-1.5B for simplicity in the following text), the performance degradation on reasoning tasks (e.g., 11.67%↓ on AIME-120 and 12.80%↓ on MATH-500) are much larger than non-reasoning tasks (e.g., 1.03%↓ on Winogrande, 3.13%↓ on Hellaswag). Similar observations can be

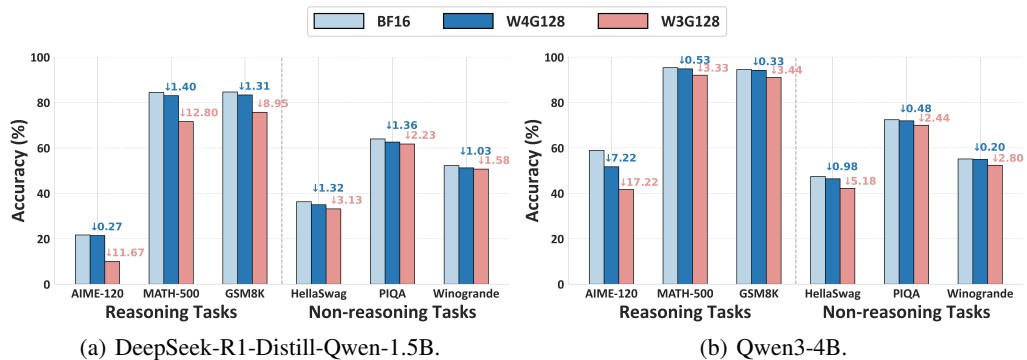

(a) DeepSeek-R1-Distill-Qwen-1.5B.  (b) Qwen3-4B.

Figure 1: The performance degradation by post-training quantization on reasoning and non-reasoning tasks. We adopt GPTQ with 3-bit weight only quantization with group size 128, and the results are based on DeepSeek-R1-Distill-Qwen-1.5B and Qwen-4B.

found for Qwen3-4B. Therefore, PTQ methods alone can hardly yield satisfactory results, which prohibit practical use of quantized reasoning models.

## 2.2 QUANTIZATION-AWARE TRAINING FOR REASONING MODELS: NEW CHALLENGES

To tackle the issue of performance degradation in PTQ, quantization-aware training (QAT) is an effective alternative. QAT simulates low-precision inference during training, allowing the model's weights to adapt to quantization and thereby preserving performance. In the forward pass, QAT inserts "fake quantization" operations to obtain $\hat{\mathbf{W}}$ or $\hat{\mathbf{X}}$ for each linear layer to quantize. Then it computes the loss objective function $\mathcal{L}(\hat{\mathbf{W}})$. In the backward pass, since the quantization function is non-differentiable, the straight-through estimator (STE) is usually adopted to allow the gradient flow back to the original weights $\mathbf{W}$, i.e., $\frac{\partial \mathcal{L}}{\partial \mathbf{W}} = \frac{\partial \mathcal{L}}{\partial \hat{\mathbf{W}}} \cdot \mathbf{1}(Q_{min} \leq \mathbf{W}/s \leq Q_{max})$. While QAT is shown to be effective on general LLMs (Liu et al., 2023; Chen et al., 2024), its benefit on reasoning models remains largely unexplored. The key challenges to consider include:

**Training Paradigm.** It remains unclear what QAT objective is preferred for continual training of reasoning models. Standard QAT simply carries over the cross-entropy objective used during full-precision pre-training or instruction fine-tuning (Liu et al., 2025b; Lee et al., 2024). Reasoning models, however, are often trained via either supervised fine-tuning with data collected from stronger teacher models (Guo et al., 2025) or reinforcement learning (Guo et al., 2025; Team et al., 2025; Yang et al., 2025). The combination of QAT with reinforcement learning is also unexplored.

**Training Overhead.** The large performance degradation by quantization usually requires intensive training for reasoning models to recover the performance. This may greatly increase the time cost of QAT, hindering the practical use when limited time is allowed for deployment.

## 3 BENCHMARKING QAT FOR REASONING MODELS

In this study, we provide a comprehensive evaluation of quantization-aware training for reasoning models. Based on the discussions in Sec. 2.2, we seek to answer the following research questions:

> **RQ1** (Sec. 3.2) How to choose the training objective of QAT for reasoning models?
> **RQ2** (Sec. 3.3) How to improve the training efficiency of QAT?
> **RQ3** (Sec. 3.4) How does QAT interact with RL algorithms such as GRPO?
> **RQ4** (Sec. 3.5) How to choose the training dataset of QAT for reasoning models?

## 3.1 SETUPS

**Quantization Settings.** We quantize all linear layers of the model, excluding the token embedding and lm_head layers. Our primary focus is on 3-bit group-wise weight-only quantization with a group-size of 128 (W3G128), for which we explore two initializations: a symmetric scheme initialized with round-to-nearest (RTN) and an asymmetric scheme initialized with GPTQ. Furthermore, we extend our approach to a joint 4-bit weight and 4-bit activation (W4A4) setting, initialized from FlatQuant, which combines per-channel symmetric weight quantization with per-token asymmetric activation quantization.

We evaluate two categories of reasoning models. For distillation-based reasoning models, we adopt DeepSeek-R1-Distill-Qwen-1.5B (Guo et al., 2025). For reasoning models obtained via reinforcement learning, we choose Qwen-3-0.6B and Qwen-3-4B (Yang et al., 2025), which are recent, high-performing open-source models.

**Datasets.** Our training pipeline consists of two phases with distinct data configurations. In the initial fine-tuning phase (SFT and KD), the weight-only setting uses the OpenR1-Math dataset (Face, 2025), which has 94k problems in its default subset, while the weight-activation setting uses 48k text sequences, each with a length of 8192, sampled from Wikitext2 (Merity et al., 2017). For the subsequent Reinforcement Learning (RL) phase, both settings use the OpenR1-Math dataset. We hypothesize that such consistency could enhance the stability of the training process.

**Evaluation Benchmarks.** We assess quantized models with different training paradigms on a range of reasoning benchmarks. These include: 1) three mathematical reasoning benchmarks sorted by difficulty: AIME-120, which consists of 120 problems from the American Invitational Mathematics Examination (AIME) from 2022 to 2025 to minimize evaluation variations; MATH-500 (Lightman et al., 2023), a benchmark containing a mix of easy and hard mathematical problems designed to test comprehensive reasoning abilities; and GSM8K (Cobbe et al., 2021), a dataset of primary school-level questions focused on basic arithmetic and algebra; 2) LiveCodeBench (Jain et al., 2024), a benchmark for evaluating large language models on code generation tasks; and 3) GPQA-Diamond (Rein et al., 2024), a graduate-level proof question and answer benchmark that tests the ability of models to generate accurate mathematical proofs. All evaluations are based on Lighteval (Fourrier et al., 2023) with the vLLM (Kwon et al., 2023) backend. The sampling temperature is set to 0.6 and top-p is fixed to 0.95. The maximum number of generated tokens is 32,768. To account for randomness, we use three seeds for each evaluation and report the average score.

**Training Implementations.** We implement and evaluate three distinct fine-tuning algorithms within our Quantization-Aware Training (QAT) framework. For Supervised Fine-Tuning (SFT), we adopt the standard cross-entropy loss objective. For Knowledge Distillation (KD), the QAT model serves as the student and is trained to mimic the output distribution of the full-precision teacher model by minimizing the KL divergence loss. For Reinforcement Learning (RL), we employ the Group Relative Policy Optimization (GRPO) algorithm (Shao et al., 2024). Further details on hyperparameters for each method are provided in Appendix A.

## 3.2 QAT TRAINING OBJECTIVES: SFT OR KD?

We aim to identify the proper training objectives for QAT. Recall that existing reasoning models are optimized via either supervised fine-tuning (e.g., R1-Qwen-1.5B) or reinforcement learning (e.g., Qwen3-4B). We first study two widely used QAT objectives (i.e., SFT and KD) for both kinds of reasoning models to investigate whether these objectives are compatible with the models' original training paradigms. We adopt weight-only quantization of W3G128 as the default setting.

Table 1 shows the performance recovery of KD and SFT for R1-Qwen-1.5B and Qwen3-4B on four reasoning benchmarks. The key observations can be summarized as follows: 1) KD demonstrates higher accuracy than SFT on reasoning models originally trained via SFT and RL. Specifically, SFT suffers an average accuracy drop of 10.51%↓ and 29.85%↓ on R1-Qwen-1.5B and Qwen3-4B, respectively. In contrast, KD results in drops of only 8.06%↓ and 9.26%↓ for the two models; 2) KD exhibits stronger synergy with both SFT- and RL-trained models, i.e., the performance drops on both models are similar (8.06%↓ vs. 9.26%↓). However, while SFT suffers a relatively moderate drop

| Model | Setting | Method | AIME120 | MATH-500 | GSM8K | AVG | Drop↓ |
|---|---|---|---|---|---|---|---|
| R1-Qwen-1.5B | BF16 | - | 21.67 | 84.4 | 84.61 | 63.56 | – |
| | W3G128 | RTN | 0.83 | 15.00 | 15.39 | 10.41 | 53.15%↓ |
| | | SFT | 10.00 | 73.60 | 75.54 | 53.05 | 10.51%↓ |
| | | KD | 14.44 | 76.20 | 75.87 | 55.50 | 8.06%↓ |
| Qwen3-4B | BF16 | - | 58.89 | 95.33 | 94.49 | 82.90 | – |
| | W3G128 | RTN | 0.00 | 1.40 | 0.99 | 0.80 | 82.10%↓ |
| | | SFT | 14.44 | 81.80 | 88.25 | 53.05 | 29.85 %↓ |
| | | KD | 37.50 | 92.00 | 91.43 | 73.64 | 9.26%↓ |

Table 1: Comparison of QAT objectives (SFT and KD) on two representative reasoning models trained either with SFT or RL, i.e., R1-Qwen-1.5B and Qwen3-4B.

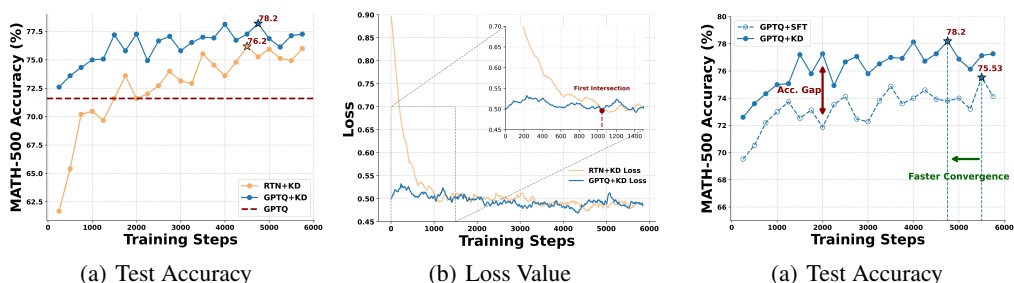

(a) Test Accuracy      (b) Loss Value      (a) Test Accuracy

Figure 2: (a) Test accuracy (%) of RTN+KD and GPTQ+KD on MATH-500. We evaluate the two methods on MATH-500. (b) Training loss of RTN+KD and GPTQ+KD.

Figure 3: Test accuracy (%) of GPTQ+KD and GPTQ+SFT on MATH-500.

on R1-Qwen-1.5B, it suffers a significantly larger performance drop (29.85%↓) on Qwen3-4B. *We therefore recommend KD over SFT as the QAT training objective, given its superior performance and stronger synergy with reasoning models originally trained with both SFT and RL.*

### 3.3 TRAINING EFFICIENCY OF QAT

**Initializing QAT with PTQ.** In previous work, it has been standard practice to initialize QAT from a pretrained full-precision model (Liu et al., 2023; Du et al., 2024). Here, we systematically investigate how PTQ-based initializations affect the convergence and accuracy of QAT. Specifically, we employ GPTQ (Frantar et al., 2022a) for initialization, using weights that have been adjusted via Hessian-based compensation prior to the quantization process. As shown in Figure 2(a)-(b), we compare the test accuracy and training loss of RTN+KD and GPTQ+KD on the MATH-500 benchmark using the R1-Qwen-1.5B model. Armed with the GPTQ-initialized weights, GPTQ+KD enjoys a higher starting point (higher test accuracy and lower loss). Besides, GPTQ+KD consistently outperforms RTN+KD and exhibits a faster convergence rate within the same number of training steps. Therefore, *PTQ acts as an effective initialization to improve the training efficiency of QAT.*

**Training Efficiency of KD.** In addition to studying the initialization of quantized models, we further compare the training efficiency of KD versus SFT, as shown in Figure 3. *The results show that KD consistently achieves higher accuracy than SFT and also converges faster.*

### 3.4 QAT WITH REINFORCEMENT LEARNING

**Prerequisites of QAT with RL.** Reinforcement Learning (RL) has demonstrated notable success in enhancing the reasoning capability of large foundation models. However, its applicability to QAT remains largely unexplored. Here, our results reveal the critical prerequisite for employing RL in QAT: *RL must be applied to a properly initialized model; otherwise, the training will collapse.* We thus study the necessary conditions that enable the integration of RL with QAT. We compare two

| RTN | KD | GRPO | AIME120 | MATH-500 | GSM8K | AVG |
|:---:|:---:|:---:|:---:|:---:|:---:|:---:|
| - | - | - | 21.67 | 84.40 | 84.61 | 63.56 |
| ✓ | - | - | 0.83 | 15.00 | 15.39 | 10.41 |
| ✓ | - | ✓ | 1.67 | 15.33 | 15.52 | 10.84 |
| ✓ | ✓ | ✓ | 14.44 | 78.00 | 77.93 | 56.79 |

Table 2: Preliminary ablations of QAT with RL based on R1-Qwen-1.5B.

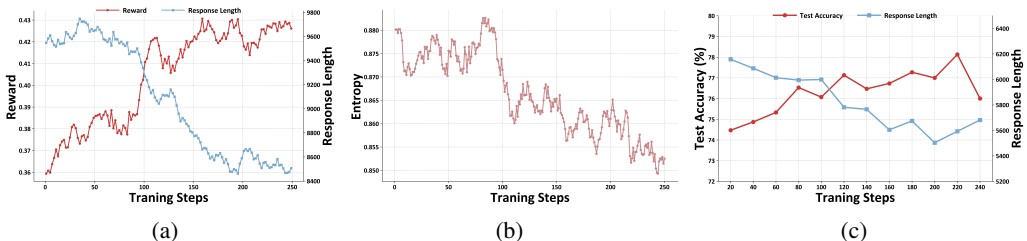

(a)  (b)  (c)

Figure 4: (a) Training reward and response length for RL QAT after cold-start. (b) Training entropy curve. (c) Test accuracy curve and corresponding response length across training steps.

settings: zero-RL QAT and cold-start RL QAT. For zero-RL QAT, we apply RL directly to the base model without any supervised fine-tuning (Guo et al., 2025), i.e., using a model pre-quantized with RTN. For cold-start RL QAT, we use a model fine-tuned with KD as the initial RL actor. During the RL phase, we employ GRPO (Guo et al., 2025) with only a correctness reward to encourage the model to sample high-reward outputs. As shown in Table 2, zero-RL QAT collapses completely, whereas cold-start RL QAT further improves reasoning ability, yielding an accuracy improvement of around 46% over the zero-RL setting. Therefore, when starting from an RTN-based model that suffers from drastic quantization error, the RL process struggles to produce high-quality outputs during inference and is unable to generate effective rewards. This contrast highlights the key prerequisite: *RL alone cannot rescue a heavily quantized model, but with a cold-start initialization, RL becomes an effective driver for enhancing reasoning ability.*

**Critical Roles of RL.** The experimental results in Figure 4 clearly demonstrate the indispensable roles that RL plays in the QAT process. First, as shown in Figure 4(a), RL simultaneously increases reward and suppresses excessive response length. This means that RL can prevent the quantized model from using response length to gain reward, instead guiding it toward truly high-quality outputs. Second, as shown in Figure 4(b), RL drives a decrease in entropy, which reduces prediction randomness and enforces more deterministic and reliable outputs. This curve indicates the role of RL in avoiding collapse while ensuring stable convergence in the presence of quantization errors. Lastly, in Figure 4(c), RL evidently improves test accuracy while reducing response length, demonstrating its ability to enhance model generalization without relying on unnecessarily verbose outputs. Overall, these results and corresponding analyses confirm the critical roles of RL in quantized reasoning models.

## 3.5 THE CHOICE OF QAT TRAINING DATASET

The choice of a QAT training dataset for reasoning models is another open challenge. In particular, it remains unclear how data from different domains influence the optimization dynamics and final performance of QAT. To study this, we compare two datasets: Wikitext2 (a natural language dataset) and OpenR1-Math (a reasoning-based math dataset). Following the setup in Section 3.3, we initialize the QAT models using PTQ. Calibration is performed using either Wikitext2 or NuminaMath-1.5, where the latter is closely aligned with OpenR1-Math[1]. Starting from these PTQ-initialized weights, we then conduct knowledge distillation for QAT on both Wikitext2 and OpenR1-Math.

---

[1]Note that OpenR1-Math consists of reasoning traces generated by DeepSeek R1 for problems from NuminaMath 1.5. More details can be found at `https://huggingface.co/datasets/open-r1/OpenR1-Math-220k`.

| Setting | PTQ | QAT | Accuracy step 1000 | step 2000 | step 3000 | Best Acc. |
|---------|-----|-----|------|------|------|------|
| W3G128 | NuminaMath-1.5 | Wikitext2 | 74.33 | 73.33 | 73.33 | 75.33 |
|  |  | OpenR1 | 75.00 | 77.27 | 77.07 | 78.53 |
| W4A4KV4 | Wikitext2 | Wikitext2 | 71.67 | 73.13 | 71.80 | 73.20 |
|  |  | OpenR1 | 43.27 | 44.40 | 43.70 | 45.80 |

Table 3: Comparison with different QAT data. With two different training datasets on the R1-Qwen-1.5B model, we present the MATH-500 test accuracy during training and the final best accuracy.

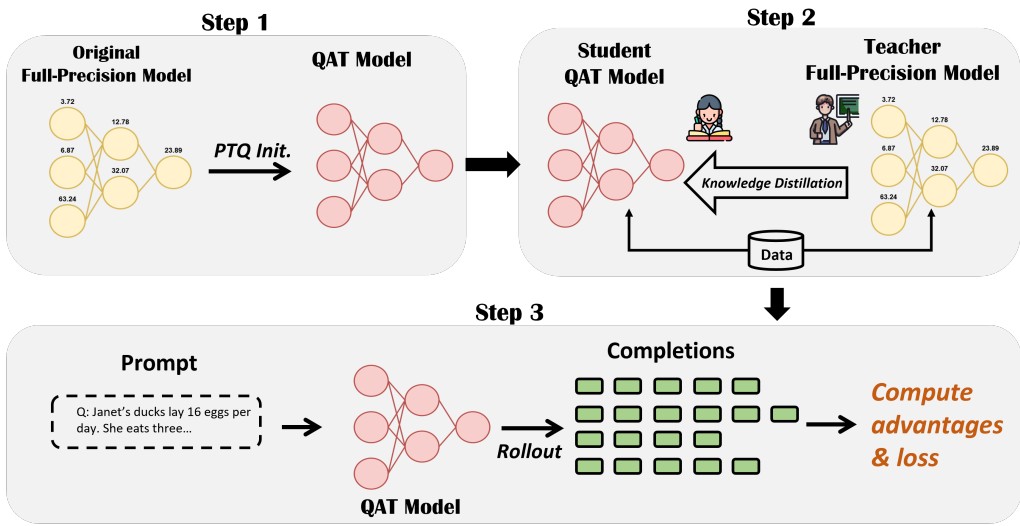

Figure 5: The overall workflow of the proposed Reasoning-QAT. Step 1: PTQ-based initialization can provide a better starting point. Step 2: KD from the original full-precision model to align the teacher's behavior, and also serve as a cold-start for subsequent RL. Step 3: Based on cold-start, RL can further recover the reasoning ability of the QAT model.

As summarized in Table 3, the results demonstrate that aligning the QAT training dataset with the PTQ calibration data leads to better performance. When QAT is performed on OpenR1-Math with PTQ calibrated on NuminaMath-1.5, the model achieves the highest accuracy (i.e., 78.53%) and exhibits faster convergence compared to the mismatched setting. Conversely, when QAT training is performed on Wikitext2 while PTQ calibration is done on NuminaMath-1.5, model performance drops clearly. These findings suggest that *it is beneficial to keep the consistency between PTQ calibration data and training dataset of QAT for quantized reasoning models, and offer practical guidance for QAT data selection for those reasoning models quantized by PTQ.*

For QAT with RL, we omit the comparisons but consistently choose OpenR1-Math as the training dataset, as it remains unclear to reward the general Wikitext2 dataset.

## 4 QAT FOR REASONING MODELS: THE ULTIMATE WORKFLOW

Based on the observations in Section 3, we provide the ultimate workflow in Figure 5, which includes three key steps to guide practical applications and support downstream usage. Note that our workflow is agonistic to the choice of quantization configurations, which can be applied to both weight-only and weight-activation quantization.

### 4.1 DETAILED PIPELINE AND ALGORITHM

• **Step 1: PTQ-based Initialization**. Motivated by Section 3.3, we rectify the latent weights with PTQ techniques as the initial state for QAT. While the QAT model still remains with continuous

| Model | W-Bits | Methods | AIME-120 | MATH-500 | GSM8K | GPQA-Diamond | LiveCode-Bench | Avg. | Drop ↓ |
|---|---|---|---|---|---|---|---|---|---|
| Qwen3-0.6B | BF16 | - | 11.11 | 74.00 | 79.00 | 28.45 | 12.94 | 41.10 | - |
| | W3G128 | RTN | 0.00 | 0.80 | 0.30 | 24.24 | 0.00 | 5.07 | -36.03 |
| | | GPTQ | 0.83 | 11.80 | 20.24 | 24.24 | 0.00 | 11.42 | -29.62 |
| | | AWQ | 0.00 | 5.20 | 10.01 | 26.77 | 0.00 | 8.40 | -32.70 |
| | | Reasoning-QAT | 3.89 | 57.80 | 67.02 | 27.78 | 1.87 | 31.67 | -9.43 |
| | W4A4KV4 | QuaRot | 0.00 | 0.00 | 0.00 | 24.24 | 0.00 | 4.84 | -36.26 |
| | | FlatQuant | 0.28 | 21.67 | 33.06 | 29.80 | 1.87 | 17.34 | -23.76 |
| | | Reasoning-QAT | 0.00 | 30.27 | 48.62 | 26.94 | 1.37 | 21.44 | -19.66 |
| R1-1.5B | BF16 | - | 21.67 | 84.40 | 84.61 | 36.87 | 16.04 | 48.72 | - |
| | W3G128 | RTN | 0.83 | 15.00 | 15.39 | 19.19 | 0.00 | 10.08 | -38.64 |
| | | GPTQ | 10.00 | 71.60 | 75.66 | 23.74 | 9.33 | 38.07 | -10.65 |
| | | AWQ | 3.33 | 48.80 | 65.81 | 37.88 | 4.85 | 32.13 | -16.58 |
| | | Reasoning-QAT | 16.39 | 79.80 | 79.35 | 30.30 | 8.21 | 42.81 | -5.91 |
| | W4A4KV4 | QuaRot | 0.00 | 1.20 | 0.76 | 8.59 | 0.00 | 2.11 | -46.61 |
| | | FlatQuant | 10.00 | 64.80 | 78.62 | 31.82 | 6.72 | 38.39 | -10.33 |
| | | Reasoning-QAT | 12.50 | 73.20 | 77.94 | 32.83 | 10.07 | 41.31 | -7.41 |
| Qwen3-4B | BF16 | - | 58.89 | 95.33 | 94.49 | 56.06 | 48.38 | 70.63 | - |
| | W3G128 | RTN | 0.00 | 1.40 | 0.99 | 10.60 | 0.00 | 2.60 | -68.03 |
| | | GPTQ | 41.67 | 92.00 | 91.05 | 41.41 | 25.00 | 58.23 | -12.4 |
| | | AWQ | 25.00 | 87.00 | 90.07 | 37.88 | 19.03 | 51.80 | -18.83 |
| | | Reasoning-QAT | 41.11 | 93.47 | 93.48 | 45.79 | 38.06 | 62.38 | -8.25 |
| | W4A4KV4 | FlatQuant | 32.78 | 89.93 | 92.12 | 47.47 | 29.10 | 58.28 | -12.35 |
| | | Reasoning-QAT | 36.67 | 91.40 | 92.42 | 48.48 | 34.95 | 60.78 | -9.85 |

Table 4: Main results of Reasoning-QAT on Qwen3-0.6B, R1-Qwen-1.5B and Qwen3-4B across various reasoning benchmarks.

weights, this initialization strategy improves its tolerance to quantization and provides a better starting point for subsequent training.

• **Step 2: Knowledge Distillation**. Building upon the model from Step 1, we perform knowledge distillation from the original full-precision model. Guided by the findings in Section 3.2, this step fine-tunes the QAT model to align its output distribution with that of the full-precision model. After that, the distilled model not only recovers from the quantization-induced degradation, but also serves as a stable cold-start actor for RL.

• **Step 3: Cold-start RL**. Following the prerequisites discussed in Section 3.4, we apply RL on top of the knowledge-distilled model from Step 2. Here, we employ GRPO (Guo et al., 2025) as the RL paradigm. This cold start design avoids the collapse issue observed when directly using RL on heavily quantized models, while utilizing the stabilized initialization to ensure reliable optimization. During this stage, RL progressively enhances the reasoning capability of the quantized model, driving more deterministic outputs and reducing randomness.

## 4.2 EMPIRICAL EVALUATIONS

**Weight-only Quantization.** We first analyze the results under the W3G128 quantization setting. The experimental results of comparison with PTQ are provided in Table 4. Across all three model scales (Qwen3-0.6B, R1-Qwen-1.5B, and Qwen3-4B), PTQ methods such as RTN, GPTQ, and AWQ show severe degradation on evaluation benchmarks, with performance drop often exceeding 30% on average. In contrast, our Reasoning-QAT consistently achieves clear accuracy recovery. For example, on Qwen3-0.6B, the average score improves from 11.42% (GPTQ) to 31.67% (Reasoning-QAT). This reduces the performance gap to full precision (BF16) by more than 20 points. Similar trends are observed for R1-Qwen-1.5B and Qwen3-4B, where our Reasoning-QAT narrows the gap to only -5.91% and -8.20%, respectively, which obviously outperforms all PTQ baselines. These results highlight that while PTQ struggles to preserve reasoning ability at 3-bit weights, our Reasoning-QAT provides a more advanced solution to bridge the quantization gap.

**Weight-activation Quantization.** We then examine W4A4KV4 quantization as a representative configuration for weight-activation quantization. This scenario is particularly challenging since weights, activations, and KV cache are quantized to low bits. Note that we implement Reasoning-QAT in this setting by loading the transformation matrices from FlatQuant as initialization and further performing QAT. Unlike the original FlatQuant, which applies layer-wise correction in isolation, our method uses network-wise adjustments during QAT. This holistic optimization makes

| | RTN | GPTQ | SFT | KD | GRPO | AIME120 | MATH-500 | GSM8K | AVG |
|---|---|---|---|---|---|---|---|---|---|
| #0 | - | - | - | - | - | 21.67 | 84.40 | 84.61 | 63.56 |
| #1 | ✓ | - | - | - | | 0.83 | 15.00 | 15.39 | 10.41 |
| #2 | ✓ | - | ✓ | - | - | 10.00 | 73.60 | 75.54 | 53.05 |
| #3 | ✓ | - | - | ✓ | - | 14.44 | 76.20 | 75.87 | 55.50 |
| #4 | ✓ | - | - | ✓ | ✓ | 14.44 | 78.00 | 77.93 | 56.79 |
| #5 | - | ✓ | - | - | - | 10.00 | 71.60 | 75.66 | 52.42 |
| #6 | - | ✓ | ✓ | - | - | 14.17 | 75.53 | 76.12 | 55.27 |
| #7 | - | ✓ | - | ✓ | - | 13.89 | 78.20 | 77.26 | 56.45 |
| #8 | - | ✓ | - | ✓ | ✓ | **16.39** | **79.80** | **79.35** | **58.51** |

Table 5: Ablation studies of Reasoning-QAT, including the PTQ initializations (i.e., RTN and GPTQ), QAT training paradigms (i.e., SFT, KD and GRPO) based on R1-Qwen-1.5B.

the model account for the propagation of quantization errors across layers, thereby handling the accumulation of mismatches that single-layer correction cannot capture. As a result, the model can adaptively correct quantization errors in a globally consistent manner rather than relying solely on static PTQ calibration. As can be seen, PTQ baselines such as QuaRot and FlatQuant suffer from large performance decreases. Our method, however, achieves consistent improvements across all model sizes. For instance, on Qwen3-4B, Reasoning-QAT raises the average score from 58.28 (FlatQuant) to 60.78, effectively narrowing the gap to full precision and demonstrating that our method can effectively tackle the degradation in W4A4KV4 quantization scenarios.

## 4.3 ABLATION STUDY

In this ablation study, we clarify the efficacy of each Reasoning-QAT components, which are PTQ initialization, KD, and GRPO. We specifically assess the 3-bit groupwise weight-only quantization on R1-Qwen-1.5B model shown in Table 5.

**GPTQ Initialization.** To investigate the impact of different weight quantization initialization strategies on the effectiveness of Quantization-Aware Training (QAT), we present QAT models starting from RTN and GPTQ in rows 1-4 and rows 5-8, respectively. It can be found that using GPTQ for initialization yields a better starting point, resulting in an average improvement of 42.01% (row 1 vs. row 5).

**The Effect of KD.** Both SFT and KD significantly recover quantization loss. With RTN initialization, SFT yields a 42.64% improvement (row 1 vs. row 2), while KD achieves an 45.09% gain (row 1 vs. row3). Regardless of initialization, the KD approach demonstrates robustly superior performance over SFT. To be specific, KD achieves higher average accuracy than SFT by 2.45% under RTN (row 1 vs. row 2) and by 1.18% under GPTQ (row 3 vs. row 4).

**The Marginal Improvement by GRPO.** To further refine the performance of quantized models trained with knowledge distillation (KD), we integrate GRPO into the training pipeline. It can be seen that GRPO further boosts KD performance by 1.29% under RTN (row 3 vs. row 4) and 2.06% under GPTQ (row 7 vs. row 8), demonstrating its effectiveness in enhancing quantized models through policy refinement.

## 5 CONCLUSION

In this work, we presented a comprehensive benchmark of quantization-aware training (QAT) for reasoning models, revealing four key insights: knowledge distillation as a versatile objective, PTQ as an effective initialization, the feasibility of combining QAT with RL, and the importance of aligning QAT data with PTQ calibration. Building on these findings, we proposed Reasoning-QAT, a three-stage workflow that consistently outperforms PTQ baselines and significantly reduces the gap to full-precision models under extreme low-bit settings. Our study provides practical guidance for the efficient deployment of quantized reasoning LLMs.

ETHICS STATEMENT

This work benchmarks and optimizes QAT for reasoning LLMs. All experiments are conducted on publicly available datasets and open-source models, without involving any private or sensitive data. We encourage downstream applications to incorporate safeguards such as usage monitoring, content filtering, and transparency reporting. We further advocate for the ethical development and deployment of reasoning LLMs, with particular attention to fairness, robustness, and accountability.

REPRODUCIBILITY STATEMENT

Experimental settings are carefully described and listed in Appendix A. We detail the model choice, dataset usage in Section 3.1, respectively. To further ensure reproducibility, we promise to open-source both the code and model checkpoints.

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

## A   TRAINING IMPLEMENTATIONS DETAILS

We list the detailed training hyper-parameters in Tables 6 and 7.

Table 6: Hyperparameters for Phase 1 (Cold Start). This phase involves Supervised Fine-Tuning (SFT) and Knowledge Distillation (KD) for models under different quantization and initialization schemes.

| Parameter | W3g128 Setting | | | | W4A4 Setting |
| --- | --- | --- | --- | --- | --- |
| | RTN Initialization | | GPTQ Initialization | | FlatQuant Init. |
| | SFT | KD | SFT | KD | KD |
| *Optimizer Settings* | | | | | |
| Optimizer | | | Adam | | |
| Learning Rate (Peak) | 2e-5 | 2e-5 | 1e-6 | 1e-6 | 1e-6$^{*}$ |
| LR Scheduler | | | Cosine Decay | | |
| Warmup Steps | 180 | 180 | 180 | 180 | 90 |
| Adam Betas ($\beta_1, \beta_2$) | | | 0.9, 0.95 | | |
| *Training Settings* | | | | | |
| Global Batch Size | | | 32 | | |
| Gradient Accumulation | | | 4 | | |
| Training Steps | 6,000 | 6,000 | 6,000 | 6,000 | 3,000 |

\* For the W4A4 KD setting, we employed differentiated learning rates for three distinct parameter groups: [Standard model weights: 1e-6, Transformation matrix and scaling factor: 5e-5, Clipping factor: 5e-4].

Table 7: Hyperparameters for Phase 2, Reinforcement Learning via GRPO. These settings are applied to models after they have completed Phase 1.

| Parameter | Value |
| --- | --- |
| *Optimizer Settings* | |
| Optimizer | Adam |
| Learning Rate (Peak) | 5e-7 |
| LR Scheduler | Cosine Decay |
| Warmup Steps | 8 |
| Adam Betas ($\beta_1, \beta_2$) | 0.9, 0.95 |
| *Training Settings* | |
| Global Batch Size | 64 |
| Gradient Accumulation | 4 |
| Training Steps | 250 |
| *Algorithm-Specific Settings* | |
| Reward Function | Correctness Reward |
| GRPO Group Size | 8 |
| Maximum Generation Length | 32768 |

## B   THE USE OF LARGE LANGUAGE MODELS

We declare that large language models (LLMs) were employed to assist with the refinement of this manuscript, specifically, for grammar checking, language polishing, and improving the clarity and fluency of the text. Additionally, LLMs were used in a limited capacity for minor debugging and syntactic correction of code snippets included in the work.

