# OpenReview forum: "Benchmarking and Advancing Quantization-Aware Training for Reasoning Models"
_ICLR.cc/2026/Conference — ICLR 2026 Conference Withdrawn Submission_

### Official Review · Reviewer_uyks · 2025-10-24

**Soundness:** 3
**Presentation:** 3
**Contribution:** 2
**Rating:** 4
**Confidence:** 3

**Summary:**

Quantization can produce large drops in reasoning benchmark performance, particularly for small models. This work comprehensively benchmarks quantization-aware training (QAT) for reasoning models, offers some insights into best practices for QAT, and introduces Reasoning-QAT, an optimized QAT workflow tailored to reasoning models which outperforms state-of-the-art PTQ methods across multiple LLM backbones and reasoning datasets.

**Strengths:**

* The authors provide the first comprehensive benchmark of quantization-aware training, according to them, and I have no reason to doubt it. This is valuable.
* The authors provide sufficient experimental evidence to convince me that, at least under certain conditions, Reasoning-QAT improves on the baselines, and they selected strong recent baselines for comparison, good base models, good choice of benchmarks.
* The ablation Table 5 is very nice.
* I think the layout of this work is rather nice. I like the way the authors highlighted and numbered their research questions, and the esthetic effect of the colors and figure placement is good.

**Weaknesses:**

* The inconsistency of the experimental settings between figures and tables is vexing; it makes it harder to figure out what is really going on in this paper. For instance, Table 1 compares SFT and KD to an RTN baseline, but Figure 1 uses GPTQ as a point of comparison, but keeps the model architectures the same. There's no reason not to include both RTN and GPTQ in both Table 1 and Figure 1. In Table 4, Qwen3-4B W4A4KV4 is missing QuaRot (only shows FlatQuant), all other models show QuaRot for W4A4KV4. Why does Table 4 compare W4A4KV4 to W3G128 instead of comparing 4-bit to 4-bit or 3-bit to 3-bit? Is it because the authors are trying to control for the difficulty of the quantization task? This would benefit from a clearer explanation.
* The W4A4KV4 version of Reasoning-QAT is dependent on FlatQuant's specific architecture and transformation matrices. This is a potential confound, as FlatQuant is also a baseline.
* While the subject is worthy of some study, the authors may be somewhat overstating the centrality of QAT. While the research literature does show that quantization can cause large performance drops in reasoning models, there are a number of mitigating factors, including model size (Liu et al. (2025a) showed that the 32B model maintained 96.77% of baseline performance at 3-bit while the 1.5B model collapsed to 67.15%), PTQ algorithm (shown by the authors as well as others), quantization bit depth, and the nature of the reasoning task, with mathematical reasoning being particularly severely affected. I do not object to the authors studying a niche topic, but I think it is important for the authors to highlight somewhere (for instance, in an appendix Limitations section) that in fact, under many circumstances, PTQ with a reasonable compression scheme such as AWQ will perform just fine, even on reasoning tasks.
* (nit) Authors should define acronyms in their figures and tables captions, e.g. W4G128 in Figure 1
* (nit) There are too many figures and tables in the main paper, it's cluttered and it makes it hard to follow the main story. Table 2, for example, would be fine in the appendix as far as I can see, Table 5 tells a similar story but is much more comprehensive.
* (nit) Figures 2, 3 and 4 are too small and hard to read, again, because the authors are trying to cram everything into the main paper. It isn't all needed.
* (nit) Why is the workflow diagram on Page 7? Shouldn't that be towards the beginning in a methods paper?

**Questions:**

* If you used a different W4A4 PTQ method would the QAT approach transfer?
* Why not show "QAT without FlatQuant" for W4A4?

---

> ### Author Response · Authors · 2025-11-26
> **Response to Reviewer uyks (Part 1)**
>
> We thank the reviewer for the detailed feedback and for recognizing our work as a "comprehensive benchmark" with "solid experiments" and a "very nice" ablation (Table 5).
>
> **W1**: "The inconsistency of the experimental settings between figures and tables is vexing; it makes it harder to figure out what is really going on in this paper. For instance, Table 1 compares SFT and KD to an RTN baseline, but Figure 1 uses GPTQ as a point of comparison, but keeps the model architectures the same. There's no reason not to include both RTN and GPTQ in both Table 1 and Figure 1. In Table 4, Qwen3-4B W4A4KV4 is missing QuaRot (only shows FlatQuant), all other models show QuaRot for W4A4KV4. Why does Table 4 compare W4A4KV4 to W3G128 instead of comparing 4-bit to 4-bit or 3-bit to 3-bit? Is it because the authors are trying to control for the difficulty of the quantization task? This would benefit from a clearer explanation."
>
> **A1**: We apologize for the confusion and clarify the distinct design purposes of these settings:
>
> 1. **Figure 1 (Motivation) vs. Table 1 (Ablation)**: Figure 1 uses GPTQ to demonstrate that even strong SOTA baselines fail disproportionately on reasoning tasks, leading to the motivation of Reasoning-QAT. Table 1 intentionally fixes initialization to RTN to **rigorously isolate the training objective's impact**, proving KD recovers performance even from a naive starting point. We will clarify these distinct intents in the captions and add a supplementary Appendix table providing the full cross-product results (RTN/GPTQ  SFT/KD).
> 2. **Missing QuaRot on Qwen3-4B**: Excluded due to architectural constraints. The standard QuaRot kernel (and the implementation in [R1], which we use for reproduction)  relies on Hadamard transformation, requiring a hidden size of . Qwen3-4B (size 2560) is mathematically incompatible without significant kernel modification.
> 3. **Comparing W4A4KV4 vs. W3G128**: Following the presentation of [R1], we do not imply a direct bit-depth comparison. Instead, we present two parallel tracks addressing different hardware bottlenecks: **Memory-Bound** (W3G128, maximizing model size) and **Compute-Bound** (W4A4KV4, maximizing throughput). We will explicitly label these tracks in the revision.
>
> **W2**: "The W4A4KV4 version of Reasoning-QAT is dependent on FlatQuant's specific architecture and transformation matrices. This is a potential confound, as FlatQuant is also a baseline."
>
> **A2**: The reviewer questioned if reliance on FlatQuant is a confound. We clarify that in W4A4KV4 settings, severe activation outliers make the optimization landscape intractable for direct quantization. Therefore, FlatQuant (or an equivalent outlier-suppression technique like QuaRot) is a prerequisite to condition the weights/activations for stable training, rather than a confounding variable. Reasoning-QAT builds upon this necessary foundation to further recover reasoning capabilities.
>
> **W3**: "While the subject is worthy of some study, the authors may be somewhat overstating the centrality of QAT. "
>
> **A3**: We agree that large models (32B+) are robust to PTQ. However, extreme quantization (W3/W4) is crucial for **democratizing reasoning on consumer hardware**. As noted by [R1], small models (1.5B-7B) do suffer significant collapse under PTQ. Therefore, Reasoning-QAT is not a niche fix, but a critical enabler for the widespread deployment of compact reasoning models where PTQ fails.
>
> **W4**: "(nit) Authors should define acronyms in their figures and tables captions, e.g. W4G128 in Figure 1."
>
> **A4**: We will explicitly define W4G128 and all acronyms in captions.
>
> **W5**: "(nit) There are too many figures and tables in the main paper, it's cluttered and it makes it hard to follow the main story. Table 2, for example, would be fine in the appendix as far as I can see, Table 5 tells a similar story but is much more comprehensive. (nit) Why is the workflow diagram on Page 7? Shouldn't that be towards the beginning in a methods paper?"
>
> **A5**: We will move Table 2 to the Appendix to declutter the text and relocate the Workflow Diagram (Figure 5) to Page 2 for an earlier overview.
>
> **W6**: "Figures 2, 3 and 4 are too small and hard to read, again, because the authors are trying to cram everything into the main paper. It isn't all needed."
>
> **A6**: We will resize Figures 2, 3, and 4 to span the full page width.

---

> ### Author Response · Authors · 2025-11-26
> **Response to Reviewer uyks (Part 2)**
>
> **Q1**: "If you used a different W4A4 PTQ method would the QAT approach transfer?"
>
> **A7**: We applied Reasoning-QAT to QuaRot initialization (W4A4KV4). The results yield two key insights:
>
> 1. **KD Transferability**: The KD stage successfully improved performance over the PTQ baseline (e.g., MATH-500: 1.20% to 11.20%), confirming the objective is transferable.
>
> 2. **RL Prerequisites** (Validated in Section 3.4): The subsequent RL stage collapsed. This empirically validates our theory that RL requires a minimum capability threshold to generate rewardable trajectories. Since the QuaRot-based model, even after KD, remained below this threshold (due to severe outliers in this architecture), it failed to sustain RL exploration.
>
> Therefore, the QAT workflow is transferable, but the RL stage requires an initialization strong enough (like FlatQuant) to provide a viable starting policy.
>
> | Initialization | Method | AIME-120 | MATH-500 | GSM8K | GPQA-Diamond | LiveCodeBench | Status |
> | :--- | :--- | :---: | :---: | :---: | :---: | :---: | :--- |
> | **QuaRot (PTQ)** | - | 0.00% | 1.20% | 0.76% | 8.59% | 0.00% | **Collapsed** |
> | **QuaRot** | KD | 1.67% | 11.20% | 12.74% | 9.09% | 1.12% | **Improved** |
> | **QuaRot** | Reasoning-QAT | 0.00% | 3.60% | 6.07% | 2.53% | 0.00% | **Collapsed** |
> | **FlatQuant** | - | 10.00% | 64.80% | 78.62% | 31.82% | 6.72% | **Converged** |
> | **FlatQuant** | Reasoning-QAT | 12.50% | 73.20% | 77.94% | 32.83% | 10.07% | **Converged** |
>
> **Q2**: "Why not show "QAT without FlatQuant" for W4A4?"
>
> **A8**: We initially omitted this baseline because it leads to collapse. To demonstrate this empirically, we conducted the requested ablation using standard **RTN initialization** (i.e., QAT without FlatQuant's transformations).
>
> 1. **Result**: As shown below, RTN-based QAT led to **complete training collapse** (Accuracy  0%), whereas FlatQuant-initialized QAT converged successfully.
>
> 2. **Conclusion**: This confirms FlatQuant (or an equivalent outlier-suppression technique) is a **necessary precondition** for stable W4A4 training, not a confounding variable.
>
> | Initialization | Method | AIME-120 | MATH-500 | GSM8K | GPQA-Diamond | LiveCodeBench | Status |
> | :--- | :--- | :---: | :---: | :---: | :---: | :---: | :--- |
> | **RTN** | KD | 0.00% | 0.60% | 1.29% | 4.55% | 0.00% | **Collapsed** |
> | **RTN** | Reasoning-QAT | 0.00% | 0.20% | 0.37% | 1.01% | 0.00% | **Collapsed** |
> | **FlatQuant** | Reasoning-QAT | 12.50% | 73.20% | 77.94% | 32.83% | 10.07% | **Converged** |
>
> We hope these responses address your concerns and clarify the contributions of our work.
>
> ----
> [R1] Quantization hurts reasoning? an empirical study on quantized reasoning models. arXiv preprint arXiv:2504.04823, 2025.

---

### Official Review · Reviewer_sihG · 2025-11-01

**Soundness:** 3
**Presentation:** 4
**Contribution:** 4
**Rating:** 8
**Confidence:** 5

**Summary:**

The goal of this study is to analyze and improve Quantization-Aware Training (QAT)  for reasoning models. Authors first try to benchmark how much quantization hurts reasoning models in two settings W3G128 and W4A4. They systematically ablate various design choices for doing QAT and then propose a reasoning QAT recipe that minimizes quantization loss. The 4 recommendations involves
* Initialization using a Post-train Quantization (PTQ) checkpoint eg GPTQ
* doing Knowledge distillation (KD) as opposed to reasoning SFT
* QAT GRPO on top yields marginal benefit too
* Matching data domain while doing PTQ calibration in first step helps

**Strengths:**

- Very comprehensive evaluations
- Solid experiments followed by concrete recipe
- Lots of good learning from the well formulated Research Questions like impact of PTQ caliberation data, KD > SFT etc.
- Results show meaningful reduction in quantization accuracy loss for both the quantization settings
- Paper is extremely well written and a delight to read.

**Weaknesses:**

- There is rich literature on logit distillation as opposed to doing reasoning SFT by distilling from stronger teacher which is not adequately referenced
- Gains from RL are unclear and it is within error for reasoning benchmarks
- avg of 3 is not enough for evals especially for AIME25
- no experiments with larger models so unclear how results will translate.
- Authors posttraining is very math focused and not a general postraining which will target multiple domains and also chat and instruction following. Its unclear if the results will translate over as it is much more tricky to get the balance right in that scenario.

**Questions:**

- I would be curious if authors try on-policy KD where you first sample a response from the quantized student for a given prompt and then minimize KL For that prompt, response compared to the teacher. There is evidence in literature that it works better [1]

- dataset for RL, DAPO is probably better

[1]https://arxiv.org/abs/2306.13649

---

> ### Author Response · Authors · 2025-11-26
> **Response to Reviewer sihG (Part 1)**
>
> We sincerely thank the reviewer for the encouraging assessment and for highlighting our work as "comprehensive," "solid," and a "delight to read." We are particularly grateful for your recognition of our "concrete recipe". Your constructive feedback has helped us significantly strengthen the paper. Below, we address your comments regarding literature, evaluation stability, and generalizability.
>
> **W1**: "There is rich literature on logit distillation as opposed to doing reasoning SFT by distilling from stronger teacher which is not adequately referenced."
>
> **A1**: We appreciate this insight. We will expand the Related Work section to explicitly discuss the distinction between reasoning SFT and logit-based distillation. We clarify that our "KD" baseline minimizes KL divergence on teacher-forced trajectories, serving as a robust foundation for QAT.
>
> **W2**: "Gains from RL are unclear and it is within error for reasoning benchmarks."
>
> **A2**: The reviewer noted RL gains might seem marginal. We respectfully argue they are significant given the context:
> 1. **Baseline Collapse**: In W3G128, PTQ often collapses to near-zero (Table 5, #1). Any consistent recovery is a qualitative breakthrough.
>
> 2. **Consistency & Mechanism**: The improvement is consistent across all benchmarks (Table 5). For example,  combining RL yields a **2.06% increase** in average accuracy on the three datasets (Table5, #7 & #8). Notably, on MATH-500, the accuracy improves from 78.13% ($\pm$0.98) to 79.80% ($\pm$0.60). The fact that the performance gain exceeds the standard deviation confirms that this improvement is genuine and not merely statistical noise. Furthermore, Figure 4 shows RL significantly reduces entropy and response length, proving it actively "sharpens" the quantized model's reasoning beyond random error.
>
> **W3**: "avg of 3 is not enough for evals especially for AIME25."
>
> **A3**: We agree that reasoning benchmarks can have high variance. To address your concern, we re-evaluated the **AIME-120** benchmark using **10 random seeds** (instead of 3) for our proposed Reasoning-QAT (W3G128 & W4A4KV4) on the R1-Qwen-1.5B model and reported average accuracy and standard deviation.
>
> | Settings | Models | Methods | AIME120 (%) |
> | :---: | :--- | :--- | :---: |
> | W3G128 | Qwen3 0.6B | GPTQ | 0.16 ± 0.35 |
> | W3G128 | Qwen3 0.6B | **Reasoning QAT** | **5.75 ± 0.83** |
> | W3G128 | R1-Qwen-1.5B | GPTQ | 10.08 ± 2.53 |
> | W3G128 | R1-Qwen-1.5B | **Reasoning QAT** | **14.83 ± 1.75** |
> | W3G128 | Qwen3 4B | GPTQ | 33.33 ± 2.81 |
> | W3G128 | Qwen3 4B | **Reasoning QAT** | **44.92 ± 2.06** |
> | W4A4KV4 | Qwen3 0.6B | Flatquant | 2.17 ± 0.87 |
> | W4A4KV4 | Qwen3 0.6B | **Reasoning** QAT | **2.50 ± 0.81** |
> | W4A4KV4 | R1-Qwen-1.5B | Flatquant | 8.50 ± 1.61 |
> | W4A4KV4 | R1-Qwen-1.5B | **Reasoning QAT** | **12.75 ± 2.26** |
> | W4A4KV4 | Qwen3 4B | Flatquant | 43.33 ± 1.92 |
> | W4A4KV4 | Qwen3 4B | **Reasoning QAT** | **44.00 ± 2.60** |
>
> **W4**: "no experiments with larger models so unclear how results will translate."
>
> **A4**: We acknowledge this limitation. As an academic research group, we are constrained by computational resources, which limited our ability to perform QAT and RL on larger models. We will explicitly state this limitation and the extrapolation potential in the Conclusion.

---

> ### Author Response · Authors · 2025-11-26
> **Response to Reviewer sihG (Part 2)**
>
> **W5**: "Authors posttraining is very math focused and not a general postraining which will target multiple domains and also chat and instruction following. Its unclear if the results will translate over as it is much more tricky to get the balance right in that scenario."
>
> **A5**: To verify robustness on non-reasoning domains, we evaluated Reasoning-QAT on **HellaSwag** (Commonsense), **PIQA** (Physics), and **Winogrande** (Commonsense) against PTQ baselines.
>
> | Setting | Model | Method | HellaSwag | PIQA | Winogrande |
> | :---: | :--- | :--- | :---: | :---: | :---: |
> | W3G128 | Qwen3-0.6B | GPTQ (Baseline) | 29.35% | 59.36% | **50.28%** |
> | W3G128 | Qwen3-0.6B | **Reasoning-QAT** | **29.41%** | **59.79%** | 50.19% |
> | W3G128 | R1-Qwen-1.5B | GPTQ (Baseline) | 32.85% | 60.01% | **51.22%** |
> | W3G128 | R1-Qwen-1.5B | **Reasoning-QAT** | **33.96%** | **62.46%** | 50.71% |
> | W3G128 | Qwen3-4B | GPTQ (Baseline) | 42.83% | 69.10% | **53.07%** |
> | W3G128 | Qwen3-4B | **Reasoning-QAT** | **42.95%** | **70.13%** | 53.00% |
> | W4A4KV4 | Qwen3-0.6B | FlatQuant (Baseline) | 34.39% | 62.24% | 52.05% |
> | W4A4KV4 | Qwen3-0.6B | **Reasoning-QAT** | **34.42%** | **62.40%** | **52.13%** |
> | W4A4KV4 | R1-Qwen-1.5B | FlatQuant (Baseline) | 34.75% | 61.26% | **51.38%** |
> | W4A4KV4 | R1-Qwen-1.5B | **Reasoning-QAT** | **34.78%** | **62.35%** | 50.32% |
> | W4A4KV4 | Qwen3-4B | FlatQuant (Baseline) | 44.94% | 69.10% | 54.10% |
> | W4A4KV4 | Qwen3-4B | **Reasoning-QAT** | **46.03%** | **70.35%** | **54.30%** |
>
> **Analysis**:
>
> 1. **Stability vs. Gains**: We observe stability rather than the dramatic gains seen in reasoning tasks. We attribute this to our exclusive use of OpenR1-Math.
>
> 2. **Data Composition Insight**: Consistent with concurrent research trends, we find a trade-off where pure reasoning data maximizes specialized gains but plateaus on general benchmarks. We will discuss in the revision that mixing general domain data into the QAT loop is the effective strategy to simultaneously boost non-reasoning metrics.
>
> **Q1**: "I would be curious if authors try on-policy KD where you first sample a response from the quantized student for a given prompt and then minimize KL For that prompt, response compared to the teacher. "
>
> **A6**: We thank the reviewer for this insightful suggestion. While On-policy KD is effective in general distillation, we prioritized Off-policy KD (Teacher Forcing) for this QAT benchmark for two specific reasons:
>
> 1. **Stability**: Low-bit models initially generate low-quality reasoning chains due to severe quantization noise. Teacher Forcing provides the stable, ground-truth guidance necessary to "repair" the weights effectively, whereas On-policy sampling risks introducing high variance at this fragile stage.
>
> 2. **Efficiency**: A primary goal of our study is to establish an efficient workflow. On-policy KD introduces significant autoregressive generation overhead during training.
>
> We will expand the Discussion section to explicitly reference the relevant literature and analyze these trade-offs (Stability vs. Mismatch) in the context of QAT.
>
> **Q2**: "dataset for RL, DAPO is probably better."
>
> **A7**: Thanks for the comment. Due to the limited time and computational resources, we will take it and explore the effectiveness of DAPO in future work.

---

> ### Comment · Reviewer_sihG · 2025-11-26
>
> Thank you for your rebuttal and running some additional experiments. They should make the people better. Given my already accepted rating, I would like to maintain the score.

---

### Official Review · Reviewer_amyM · 2025-11-02

**Soundness:** 2
**Presentation:** 1
**Contribution:** 2
**Rating:** 2
**Confidence:** 4

**Summary:**

This paper investigates the performance degradation of reasoning-focused Large Language Models (LLMs) under extreme low-bit quantization. The authors find that standard Post-Training Quantization (PTQ) methods are insufficient for these models. They propose Reasoning-QAT, a three-stage workflow combining PTQ-based initialization, knowledge distillation (KD), and a "cold-start" reinforcement learning (RL) phase. Experiments show this method significantly recovers performance on several reasoning benchmarks compared to PTQ baselines.

**Strengths:**

Addresses a Significant Problem: The work tackles the critical and timely challenge of deploying reasoning models efficiently, focusing on extreme low-bit quantization where existing methods fail.

Strong Empirical Validation: The proposed Reasoning-QAT workflow is shown to be effective and robust, consistently outperforming strong PTQ baselines across some models and benchmarks.

Provides Actionable Insights: The paper produces practical guidelines, such as the superiority of KD for QAT and the critical need for a "cold-start" before applying RL, which are of immediate use to practitioners.

**Weaknesses:**

Limited Novelty: The core weakness is the lack of algorithmic innovation. The proposed workflow is a pipeline of existing techniques (QAT, KD, RL) applied to a new domain. While the execution is effective, it does not introduce a new fundamental concept.

Overstated Contribution: The paper's claim to be a "comprehensive benchmark" is not supported by the limited scope of the experiments, which explore only one RL algorithm and a single pipeline configuration. This overclaiming undermines the paper's credibility.

Lack of Mechanistic Insight: The paper successfully shows what works but provides little explanation for why. For example, it does not offer a deep analysis of why SFT fails catastrophically on RL-trained models or explore the potential dual role of quantization noise as a regularizer in the RL phase.

Ambiguous Framing: The paper's narrative is caught between being a "benchmark" paper and a "novel method" paper. This dual identity weakens its focus. If the primary goal is to be a benchmark, the scope needs to be broader. If it is to propose a new method, the novelty needs to be more clearly articulated against prior art, and the analysis needs to be deeper.

**Questions:**

Clarification of Contribution: Is the primary contribution intended to be the Reasoning-QAT workflow or the benchmark itself? If it is a benchmark, can you justify the exclusion of other common RL algorithms and QAT methods?

Mechanism of SFT Failure: Do you have a hypothesis for the severe performance degradation when applying SFT to the RL-trained model? Could this be a form of catastrophic forgetting of the RL policy?

Role of Quantization Noise: The work treats quantization noise as purely detrimental. Did you consider that this noise might serve as a form of exploration-enhancing regularization during the RL phase, as suggested by other recent work like QERL?

---

> ### Author Response · Authors · 2025-11-26
> **Response to Reviewer amyM (Part 1)**
>
> We thank the reviewer for the rigorous assessment and for acknowledging our work's "strong empirical validation" and "actionable insights." We appreciate the opportunity to clarify our contribution scope and address the mechanistic questions regarding RL dynamics and quantization noise.
>
> **W1**: "Limited Novelty: The core weakness is the lack of algorithmic innovation. The proposed workflow is a pipeline of existing techniques (QAT, KD, RL) applied to a new domain. While the execution is effective, it does not introduce a new fundamental concept."
>
> **A1**: The reviewer noted that the workflow combines existing techniques without fundamental algorithmic innovation.
>
> 1. **Benchmark vs. Algorithm**: We respectfully clarify that our primary contribution is positioned as a **systematic benchmark study**, as stated in the title. Our goal is not to invent a new quantization algorithm, but to **establish better understanding** of traditional QAT methods in the emerging era of reasoning models.
> 2. **Systematic Discovery, Not Arbitrary Stacking**: We argue that the proposed workflow is not a trivial combination. As evidenced by our experiments, naive application of these techniques often leads to failure.
>
> 	a. **Discovery of Failure Modes**: We identified distinct failure modes specific to reasoning models, including: (1) "Improper Training Objective", where standard SFT fails to recover accuracy (see Table 1); (2) "RL Collapse", where Zero-RL QAT suffers a catastrophic drop to 1.67% accuracy (see Table 2); and (3) "Improper Data Choice", where inconsistent data domains lead to significant performance degradation (see Table 3).
>
> 	b. **Establishing Best Practices**: Our benchmark leads to **helpful findings** (e.g., PTQ Initialization and KD-based Cold-start RL) in practice that how to quantize reasoning models in low bits. We believe establishing this rigorous "Practice" workflow is a critical contribution that saves the community from "trial-and-error" pitfalls.
>
> **W2**: "Overstated Contribution: The paper's claim to be a "comprehensive benchmark" is not supported by the limited scope of the experiments, which explore only one RL algorithm and a single pipeline configuration. This overclaiming undermines the paper's credibility."
>
> **A2**: The reviewer argued that our "comprehensive" claim is unsupported because we explore "only one RL algorithm." We respectfully contend that this critique stems from a fundamental misunderstanding of the paper's intended scope. **We must clarify: This is a benchmark of Quantization-Aware Training (QAT) strategies, NOT a benchmark of Reinforcement Learning (RL) algorithms in QAT.**
>
> 1. **Defining the Scope**: The core research question is how to adapt quantization methodologies (Initialization, Data, Objectives) to the reasoning era. Evaluating the variance between different RL optimizers (e.g., PPO vs. GRPO) is orthogonal to this goal and explicitly outside our scope.
> 2. **"Comprehensiveness" in the QAT Domain**: Our study is comprehensive within the relevant QAT design space, covering dimensions previously unexplored for reasoning models: (1) Initialization (RTN vs. PTQ); (2) Training Objectives (SFT vs. KD vs. RL); (3) Quantization Regimes (Memory-Bound W3 vs. Compute-Bound W4), and (4) Data Domain Consistency.
> 3. **Correction on "Single Pipeline"**: The reviewer's claim that we explored "only a single pipeline configuration" is factually incorrect. As explicitly shown in Table 5, we systematically constructed and compared multiple distinct pipelines (e.g., RTN$\rightarrow$SFT/KD,  GPTQ$\rightarrow$SFT/KD, GPTQ$\rightarrow$KD$\rightarrow$RL) to empirically derive the optimal solution. We did not arbitrarily "execute" one workflow; "Reasoning-QAT" is the empirically proven winner emerging from this rigorous search space.

---

> ### Author Response · Authors · 2025-11-26
> **Response to Reviewer amyM (Part 2)**
>
> **W3**: "Lack of Mechanistic Insight: The paper successfully shows what works but provides little explanation for why. "
>
> **A3**: To address your concern, we summarize the specific reasons why each module works, supported by our experimental evidence:
>
> 1. **Why PTQ Initialization Works**: As analyzed in Section 3.3 and Figure 2, PTQ provides a better starting point (lower loss, higher accuracy) . This allows the model to start training from a state that is already more tolerant to quantization noise than standard RTN.
>
> 2. **Why KD Works**: As shown in Table 1, KD serves as a robust objective that aligns the quantized student's output distribution with the full-precision teacher. This "soft" target is easier to optimize than the "hard" SFT label for a quantized model, effectively preventing the performance degradation observed in SFT-only QAT.
>
> 3. **Why RL Works**: As detailed in Section 3.4 and Figure 4, our analysis reveals two mechanisms:
>
> 	a. **Entropy Reduction**: RL drives a decrease in prediction entropy (Figure 4b, Line 306), forcing the model to be more deterministic and reliable, which directly counters the uncertainty introduced by quantization errors.
>
> 	b. **Length Suppression**: RL prevents the model from "hacking" the reward with verbose, meaningless tokens (Figure 4a, Figure 4c), guiding it back to valid reasoning paths.
>
> **W4**: "Ambiguous Framing: The paper's narrative is caught between being a "benchmark" paper and a "novel method" paper. "
>
> **A4**: The reviewer noted a dual identity. We explicitly frame this work as a **Systematic Benchmark Study**. The "Reasoning-QAT" workflow is not an arbitrary novelty but the **"Best Practice" outcome** derived from our benchmark results. Our contribution lies in identifying unique failure modes (e.g., RL Collapse) and validating the necessary workflow to overcome them.

---

> ### Author Response · Authors · 2025-11-26
> **Response to Reviewer amyM (Part 3)**
>
> **Q1**: "Clarification of Contribution: Is the primary contribution intended to be the Reasoning-QAT workflow or the benchmark itself? If it is a benchmark, can you justify the exclusion of other common RL algorithms and QAT methods?"
>
> **A1.1**: The reviewer asked to justify the exclusion of other RL algorithms. Our decision to fix the RL component to **GRPO** is driven by two strategic factors:
>
> 1. **Algorithmic Consistency**: The base models benchmarked (DeepSeek-R1, Qwen3) were originally trained using GRPO. Using the same algorithm for QAT ensures we simulate the model's native training dynamics in the quantized environment. This allows us to verify if reasoning capabilities can be recovered using the original optimization method, without the interference of shifting to a different RL paradigm.
>
> 2. **Research Scope (QAT vs. RL Benchmark)**: As clarified in our Response to Weakness 2, the primary objective of this study is to benchmark Quantization-Aware Training (QAT) strategies (isolating variables like Initialization, Data Domain, and Objectives), not to conduct a comparative study of RL optimizers. Evaluating the variance between different RL algorithms (e.g., PPO vs. GRPO) is orthogonal to our goal. Introducing them would create confounding variables, obscuring our core investigation into how quantization specific factors affect reasoning recovery.
>
> **A1.2**: The reviewer correctly noted that we exclude other QAT methods. We address this in two parts:
>
> **First, Benchmarking against Standard QAT**:
>
> We clarify that Table 5 (Rows 1-4) provides a rigorous comparison against standard QAT paradigms. Specifically, we adopt **Cross-Entropy loss** [R1] as the standard SFT-QAT baseline and **KL-Divergence** [R2] as the KD-QAT baseline. Our results demonstrate that our proposed approach significantly outperforms both: surpassing SFT-QAT by **+5.46%** (58.51% vs. 53.05%) and KD-QAT by **+3.01%** (58.51% vs. 55.50%) in average accuracy.
>
> **Second, Comparison with Specialized SOTA QAT Methods**:
>
> To further address your concern, we have conducted additional experiments comparing our workflow with two recent state-of-the-art QAT methods: **BitDistiller** [R3] and **EfficientQAT** [R4]. Since these methods were originally designed for general-purpose LLMs (e.g., Llama) and non-reasoning tasks, simply quoting their reported numbers would be invalid. To ensure a fair, apples-to-apples comparison, we reproduced both algorithms using the same OpenR1-Math dataset and the same base model (R1-Qwen-1.5B) used in our work.
>
> In summary, the results in the following table demonstrate that Reasoning-QAT consistently outperforms both the standard SFT baseline and adapted SOTA QAT methods in the reasoning context. This confirms that our specific workflow (especially the inclusion of Cold-start RL) is essential for maximizing performance in reasoning tasks.
>
> We respectfully highlight that, to the best of our knowledge, **our work is the first to systematically explore QAT specifically for reasoning models**. Consequently, there were no pre-existing baselines in the literature targeting this specific domain-intersection (QAT + Reasoning) prior to this study.
>
> **New Comparative Results**:
> | Method | Description | AIME120 | MATH-500 | GSM8K | AVG |
> | :--- | :--- | :---: | :---: | :---: | :---: |
> | **Standard SFT-QAT** | Standard Cross-Entropy QAT | 10.00% | 73.60% | 75.54% | 53.05% |
> | **EfficientQAT** | Efficient QAT with Block-wise reconstruction | 10.83% | 74.20% | 76.26% | 53.76% |
> | **BitDistiller** | Self-Distillation QAT | 14.72% | 78.00% | 78.46% | 57.06% |
> | **Reasoning-QAT** | Ours (PTQ Init + KD + RL) | **16.39%** | **79.80%** | **79.35%** | **58.51%** |
>
> The results demonstrate that Reasoning-QAT consistently outperforms both the standard SFT baseline and adapted SOTA QAT methods in the reasoning context. This confirms that our specific workflow (especially the inclusion of Cold-start RL) is essential for maximizing performance in reasoning tasks.

---

> ### Author Response · Authors · 2025-11-26
> **Response to Reviewer amyM (Part 4)**
>
> **Q2**: "Do you have a hypothesis for the severe performance degradation when applying SFT to the RL-trained model? Could this be a form of catastrophic forgetting of the RL policy?"
>
> **A2**: We respectfully clarify a misunderstanding: **SFT does NOT degrade performance compared to the quantized initialization**. As shown in Table 1, SFT significantly improves accuracy over RTN (e.g., 0.80%  53.05% on Qwen3-4B), refuting the "catastrophic forgetting" hypothesis regarding capability loss. KD outperforms SFT because soft targets (logits) preserve the teacher's uncertainty ("dark knowledge"), allowing faithful recovery of the delicate reasoning policy, whereas SFT's hard labels discard this probabilistic nuance.
>
> **Q3**: "Role of Quantization Noise: The work treats quantization noise as purely detrimental. Did you consider that this noise might serve as a form of exploration-enhancing regularization during the RL phase, as suggested by other recent work like QERL?"
>
> **A3**: While QeRL finds noise beneficial in **4-bit LoRA**, we argue this is **regime-dependent**. In our **extreme low-bit (W3)** reasoning context, noise is destructive:
>
> 1. **Threshold & Fragility**: The magnitude of 3-bit noise is exponentially higher than 4-bit LoRA, overwhelming the signal and breaking the fragile logical chains required for reasoning.
>
> 2. **Evidence of Collapse**: Our Table 2 shows that "**Zero-RL QAT**" (RL on quantized weights without KD) leads to **complete collapse (1.67%)**. If noise were a regularizer here, this setting should have worked. This confirms that in this regime, noise destroys the policy, necessitating the "Cold-start" (KD) phase.
>
> ----
>
> [R1] Q-bert: Hessian based ultra low precision quantization of bert. AAAI 2020.
> [R2] Llm-qat: Data-free quantization aware training for large language models. ACL Findings 2024.
> [R3] Bitdistiller: Unleashing the potential of sub-4-bit llms via self-distillation. arXiv preprint arXiv:2402.10631, 2024.
> [R4] Efficientqat: Efficient quantization-aware training for large language models. ACL 2025.

---

### Official Review · Reviewer_EgAz · 2025-11-04

**Soundness:** 2
**Presentation:** 3
**Contribution:** 2
**Rating:** 2
**Confidence:** 5

**Summary:**

This provides a systematic study of applying QAT to reasoning large language models such as DeepSeek-R1 and Qwen. It benchmarks QAT under different training objectives, initialization methods, reinforcement learning integration, and data choices. Based on their findings, they propose Reasoning-QAT, a three-stage workflow that combines PTQ initialization, knowledge distillation, and cold-start reinforcement learning. Experiments show that Reasoning-QAT consistently outperforms state-of-the-art PTQ baselines.

**Strengths:**

1) The paper studies an increasingly important problem of applying QAT to LLMs and provides an empirical benchmark.
2) The paper is generally well-written with clear organization.

**Weaknesses:**

1) The paper’s originality is limited. The proposed “Reasoning-QAT” pipeline merely stacks existing techniques (PTQ warm-start, KD, and RL fine-tuning) without introducing any new algorithmic innovation, theoretical formulation, or quantization method.
2) The authors report accuracy improvements but offer little explanation for why the observed effects occur. For example, there is no detailed investigation into the interaction between quantization noise, KD signal, and RL reward dynamics.
3) It will be better if authors can investigate more architectures besides DeepSeek/Qwen-family models.
4) The authors benchmark various QAT methods; however, the proposed Reasoning-QAT is primarily compared against PTQ baselines. Since QAT inherently benefits from retraining, such a comparison is not entirely fair and weakens the convincingness of the claimed improvements.

**Questions:**

1) The proposed Reasoning-QAT mainly combines PTQ initialization, KD, and RL fine-tuning. Can the authors clarify what specific interactions or dependencies exist between these stages beyond simple sequential training?
2) Can authors compare Reasoning-QAT with SOTA QAT methods?

---

> ### Author Response · Authors · 2025-11-26
> **Response to Reviewer EgAz (Part 1)**
>
> We sincerely thank the reviewer for the detailed assessment and for recognizing our work as a "systematic study". We appreciate the opportunity to clarify our contribution scope and present new comparative results.
>
> **W1**: "The paper's originality is limited. merely stacks existing techniques (PTQ warm-start, KD, and RL fine-tuning) without introducing any new algorithmic innovation, theoretical formulation, or quantization method."
>
> **A1**: We respectfully clarify that the primary goal of this paper is to serve as a **comprehensive benchmark study**, as explicitly stated in the title and abstract. Our contribution is not to propose a single new algorithm, which we believe there are already abundant of them. Instead, we seek to establish a better understanding of how to quantize reasoning models in the practical workflow. The key findings include:
> 1. **PTQ Initialization**: We show that it is a crucial step to first apply PTQ before QAT, with the purpose of significantly accelerating QAT convergence compared to random/RTN initialization (Section 3.3).
> 2. **KD as a Prerequisite**: While LLMs are identified that the loss objective function of knowledge distillation is in general helpful for both SFT/RL trained LLMs, In particular, it acts as the necessary condition for the subsequent RL fine-tuning stage: without KD, the RL phase fails to explore valid reasoning paths (Section 3.4).
> 3. **RL for Reasoning Recovery**: We demonstrated that RL is the key to recovering reasoning capabilities by reducing the entropy introduced by quantization noise (Section 3.4).
> 4. **Consistency in Data Domain**: Our findings suggest that keep the consistency between PTQ calibration data and training dataset of QAT is beneficial for accelerating QAT convergence and helps improve final performance (Section 3.5).
>
> We believe these findings provide valuable guidance for quantizing reasoning models in practice.
>
> **W2**: "The authors report accuracy improvements but offer little explanation for why the observed effects occur. "
>
> **A2**：We apologize that our analysis regarding the underlying mechanisms might not have been explicitly emphasized in the initial submission. We will ensure to highlight these explanations in the revised version. To address your concern, we summarize the specific reasons why each module works, supported by our experimental evidence:
> 1. **Why PTQ Initialization Works**: As analyzed in Section 3.3 and Figure 2, PTQ provides a better starting point (lower loss, higher accuracy) . This allows the model to start training from a state that is already more tolerant to quantization noise than standard RTN.
> 2. **Why KD Works**: As shown in Table 1, KD serves as a robust objective that aligns the quantized student's output distribution with the full-precision teacher. This "soft" target is easier to optimize than the "hard" SFT label for a quantized model, effectively preventing the performance degradation observed in SFT-only QAT.
> 3. **Why RL works (RL Dynamics)**: As detailed in Section 3.4 and Figure 4, our analysis reveals two mechanisms:
>
> 	a. **Entropy Reduction**: RL drives a decrease in prediction entropy (Figure 4b, Line 306), forcing the model to be more deterministic and reliable, which directly counters the uncertainty introduced by quantization errors.
>
> 	b. **Length Suppression**: RL prevents the model from "hacking" the reward with verbose, meaningless tokens (Figure 4a, Figure 4c), guiding it back to valid reasoning paths.
>
> **W3**: "It will be better if authors can investigate more architectures besides DeepSeek/Qwen-family models."
>
> **A3**: We focused on **DeepSeek-R1** and **Qwen3** because they are currently the dominant open-source representatives of the two primary reasoning paradigms: **Distillation-based** (DeepSeek) and **Pure RL-based** (Qwen). We agree that experiments on more LLM architectures would be more convincing, yet due to limited time and computation resources in the rebuttal period, we leave this part as our future work.

---

> ### Author Response · Authors · 2025-11-26
> **Response to Reviewer EgAz (Part 2)**
>
> **W4 & Q2**: "The authors benchmark various QAT methods; however, the proposed Reasoning-QAT is primarily compared against PTQ baselines. Since QAT inherently benefits from retraining, such a comparison is not entirely fair and weakens the convincingness of the claimed improvements."
>
> **A4**: The reviewer correctly noted that we mainly compared with PTQ. We address this in two parts:
>
> **Benchmarking against Standard QAT**. We clarify that Table 5 (Rows 1-4) provides a rigorous comparison against standard QAT paradigms. Specifically, we adopt **Cross-Entropy loss** [R1] as the standard SFT-QAT baseline and **KL-Divergence** [R2] as the KD-QAT baseline. Our results demonstrate that our proposed approach significantly outperforms both: surpassing SFT-QAT by **+5.46%** (58.51% vs. 53.05%) and KD-QAT by **+3.01%** (58.51% vs. 55.50%) in average accuracy.
>
> **Comparison with Specialized SOTA QAT Methods**. To further address your concern, we have conducted additional experiments comparing our workflow with two recent state-of-the-art QAT methods: **BitDistiller** [R3] and **EfficientQAT** [R4]. Since these methods were originally designed for general-purpose LLMs (e.g., Llama) and non-reasoning tasks, simply quoting their reported numbers would be invalid. To ensure a fair, apples-to-apples comparison, we **reproduced** both algorithms using the same **OpenR1-Math** dataset and the same base model (**R1-Qwen-1.5B**) used in our work.
>
> In summary, the results in the following table demonstrate that Reasoning-QAT consistently outperforms both the standard SFT baseline and adapted SOTA QAT methods in the reasoning context. This confirms that our specific workflow (especially the inclusion of Cold-start RL) is essential for maximizing performance in reasoning tasks.
>
> We respectfully highlight that, to the best of our knowledge, **our work is the first to systematically explore QAT specifically for reasoning models**. Consequently, there were no pre-existing baselines in the literature targeting this specific domain-intersection (QAT + Reasoning) prior to this study.
>
> **New Comparative Results**:
>
> | Method | Description | AIME120 | MATH-500 | GSM8K | AVG |
> | :--- | :--- | :---: | :---: | :---: | :---: |
> | **Standard SFT-QAT** | Standard Cross-Entropy QAT | 10.00% | 73.60% | 75.54% | 53.05% |
> | **EfficientQAT** | Efficient QAT with Block-wise reconstruction | 10.83% | 74.20% | 76.26% | 53.76% |
> | **BitDistiller** | Self-Distillation QAT | 14.72% | 78.00% | 78.46% | 57.06% |
> | **Reasoning-QAT** | Ours (PTQ Init + KD + RL) | **16.39%** | **79.80%** | **79.35%** | **58.51%** |
>
> **Q1**: "What specific interactions or dependencies exist between these stages beyond simple sequential training?"
>
> **A5**: We did not simply stack existing techniques. Instead, we systematically validated the specific role, interaction, and necessary conditions for each component:
> - **PTQ Initialization**: We proved it is not just an option but a crucial step to provide a calibrated starting point, significantly accelerating QAT convergence compared to random/RTN initialization (Section 3.3).
> - **KD as a Prerequisite**: We identified that KD is the necessary condition for the subsequent RL stage. Without the distribution alignment provided by KD, the RL phase fails to explore valid reasoning paths (Section 3.4).
> - **RL for Reasoning Recovery**: We demonstrated that RL is the key to recovering reasoning capabilities by reducing the entropy introduced by quantization noise (Section 3.4).
>
> ----
> [R1] Q-bert: Hessian based ultra low precision quantization of bert. AAAI 2020.
> [R2] Llm-qat: Data-free quantization aware training for large language models. ACL Findings 2024.
> [R3] Bitdistiller: Unleashing the potential of sub-4-bit llms via self-distillation. arXiv preprint arXiv:2402.10631, 2024.
> [R4] Efficientqat: Efficient quantization-aware training for large language models. ACL 2025.

---

### Note · Authors · 2026-01-04

I have read and agree with the venue's withdrawal policy on behalf of myself and my co-authors.